# Biological Characteristics and Fungicide Screening of *Colletotrichum fructicola* Causing Mulberry Anthracnose

**DOI:** 10.3390/microorganisms12122386

**Published:** 2024-11-21

**Authors:** Ping Li, Xue Dai, Siyi Wang, Qian Luo, Qingqing Tang, Zijing Xu, Weiguo Zhao, Fuan Wu

**Affiliations:** 1Jiangsu Key Laboratory of Sericultural Biology and Biotechnology, School of Biotechnology, Jiangsu University of Science and Technology, Zhenjiang 212100, China; lee_ping2020@163.com (P.L.); daix1218@163.com (X.D.); 15988586341@163.com (S.W.); 17762375110@163.com (Q.L.); 13228267781@163.com (Q.T.); qq222383704@163.com (Z.X.); wgzsri@126.com (W.Z.); 2Key Laboratory of Silkworm and Mulberry Genetic Improvement, Ministry of Agriculture and Rural Affairs, The Sericultural Research Institute, Chinese Academy of Agricultural Sciences, Zhenjiang 212100, China

**Keywords:** anthracnose, *Colletotrichum fructicola*, carbendazim, propiconazole, mycelial morphology

## Abstract

Mulberry is an important economic crop in China that is widely planted and has important edible and medicinal value. Anthracnose, a critical leaf disease, severely compromises the yield and quality of mulberry trees. However, there are many kinds of pathogens causing mulberry anthracnose and it is difficult to control. This study was undertaken to elucidate the biological characteristics of *Colletotrichum fructicola*, the pathogen responsible for mulberry leaf spot in Zhejiang Province, and to screen out effective fungicides for its management. The biological characteristics of the pathogen were studied using the cross method and spore counting method, while the sensitivity of the pathogen to seven different fungicides was determined using the growth rate method. The findings indicated that potato dextrose agar (PDA) is the optimal growth medium for the pathogen. The pathogen was capable of growing across a temperature range of 5 to 40 °C, with optimum growth observed at 25 °C. Exposure to a 56 °C water bath for 10 min resulted in the death of the pathogen. It was also found to grow and sporulate within a pH range of 4 to 12, with an optimum pH of 7. Under alternating 12 h light and dark cycles, the colonies grew rapidly and produced abundant spores. Among the fungicides tested, 97% carbendazim WP exhibited the best inhibitory effect, with an EC_50_ (concentration for 50% of maximal effect) value of 0.0242 μg/mL. This was followed by 35% propiconazole SC, which had an EC_50_ of 0.4180 μg/mL. The fungicidal effect of 25 g/L fludioxonil SSCC was relatively poor, with an EC_50_ value of 103.0170 μg/mL. This study clarifies the optimal conditions for the growth and sporulation of the mulberry anthracnose pathogen and identifies fungicides with effective inhibitory properties. These findings will provide valuable guidance for field applications and disease management in controlling mulberry anthracnose.

## 1. Introduction

Mulberry (*Morus* L.) is an economically and ecologically important tree species that is widely distributed worldwide, especially in China [1,2]. It serves various purposes, including as feed for silkworms, livestock, and poultry, as well as in the production of food for humans and traditional Chinese medicine due to its high protein and active ingredients content [3]. However, the production of mulberry is threatened by numerous diseases, which limit the healthy growth and productivity of the trees [3]. Among these diseases, anthracnose, caused by *Colletotrichum* spp., is a key limiting factor for mulberry production in many growing areas in China, often leading to significant yield losses.

Anthracnose is a widespread disease that reduces crop yield and quality, resulting in significant economic losses [4]. *Colletotrichum*, the causal agent of anthracnose, is considered one of the top 10 fungal genera of economic and scientific importance [5]. Mulberry anthracnose was first reported in 1925. Hara (1954) named the pathogen *Colletotrichum morifolium* Hara, which is a common mulberry leaf disease in Japan [6,7]. Over the past 20 years, it has been believed that the period from June to September each year is the critical time for anthracnose infection of mulberry leaves. During this period, from early summer to autumn, leaves infected with this pathogen exhibit brown necrotic spots or stripes [8,9]. Subsequently, Yoshida et al. isolated anthracnose from diseased mulberry leaves in different areas of Japan. The pathogenicity was attributed to *C. dematium*, *C. gloeosporioides*, and *C. acutatum* [10,11]. The occurrence of this disease limits the productivity and quality of leaves, leading directly to significant losses in leaves used for silkworm feeding and livestock breeding [10], as well as in tea production. Yoshida and Shirata [12] have demonstrated that *C. dematium* possesses the ability to overwinter in both visibly and latently infected leaves, thereby serving as the primary source of infection the following year. However, there have been few studies on anthracnose of mulberry in China. Until 1982, Tian reported the spread of mulberry anthracnose in China. This widely spread fungal parasitic leaf disease was known to infect mulberry leaves and seedlings, primarily damaging mature leaves at the base of branches by causing withered spots or the yellowing of entire leaves, with a particularly severe impact on mulberry seedlings [13,14]. In the summer of 2010, brown to black spots were observed on the leaves of paper mulberry (*Broussonetia papyrifera* (L.) Venten) trees in Baiwangshan Forest Park in Beijing, China. After identification, the pathogen was determined to be *C. gloeosporioides* [15]. It was reported that *C. fioriniae* has the strongest infectivity to Sichuan white mulberry, leading to typical brown necrotic spots or stripes on the leaves. This was followed by *C. brevisporum*, *C. karstii*, and *C. kahawae* subsp. *Ciggaro* [16]. For the first time, *C. aenigma* was identified by analyzing the *ITS*, *GAPDH*, *ACT*, *CHS-1*, *GS*, *TUB2*, and *CAL* genes, and by combining these results with morphological features. The species was found to infect mulberry trees, causing anthracnose in Wuhan, China [17].

Although there have been reports of *C. fructicola* being isolated from diseased mulberry leaves, it is not generally considered pathogenic to mulberry trees [16]. To date, there have been no reports of mulberry anthracnose caused by this pathogen in China. In Zhejiang Province, China, the planting area of mulberry trees ranks sixth nationally, with approximately 445,000 acres under cultivation. Of this area, about 80,000 acres are planted in Qiandao Lake, Zhejiang. Anthracnose is a common and widely distributed fungal leaf disease in regional mulberry production, but the specific pathogen responsible for anthracnose in the region has not been clearly identified. In the initial phase of this study, leaf tissues from the affected area were collected, and *C. fructicola* was isolated and identified as the main pathogen causing anthracnose in the region. Nevertheless, its growth characteristics were unknown. This study measured the growth characteristics of this strain. Additionally, *C. fructicola* causing anthracnose on *Glycine max* in Brazil was reported in 2023 [18]. Similarly, *C. fructicola* can cause anthracnose in rubber trees, and its biological characteristics were investigated, including determining the optimal growth medium, temperature, light conditions, and pH value, along with measuring its indoor toxicity [19].

Mulberry anthracnose is one of the most difficult to control and highly destructive fungal diseases in the process of mulberry cultivation. The evolution of pathogenic bacteria is more diverse due to varying geographical locations and climatic conditions. In recent years, the classification requirements for anthracnose have become increasingly complex, and the symptoms of anthracnose after infecting mulberry also vary. Chemical protection is an important method for controlling plant diseases and pests in agricultural production. Currently, the prevention and control of diseases caused by anthracnose fungi still rely on chemical fungicides [20]. In China, the registered fungicides used for anthracnose prevention and control can be broadly divided into 11 categories based on their chemical structure, including imidazoles, triazoles, methoxyacrylates, dithiocarbamates, and others [21]. Studies have shown that the application of 40% Hekuling wettable powder, 70% thiophanate methyl wettable powder diluted 1000 times, and carbendazim effectively controls anthracnose [22]. The efficacy of fungicides against *C. cocodes* and *C. truncatum*, the pathogens causing fungal leaf spot disease in spinach, indicated that chlorothalonil, mancozeb, and pyraclostrobin were the most effective [23]. In the screening of fungicides for the prevention and control of anthracnose caused by *Colletotrichum acutatum* in strawberries, quinone-outside inhibitor (QoI) fungicides were found to exhibit the strongest antibacterial activity [24]. Extensive research on the efficacy of anthracnose treatment has led to the identification of several highly effective fungicides. However, different species or strains of the same *Colletotrichum* pathogen exhibit varying responses to fungicides [25]. *C. acutatum* isolates and *C. gloeosporioides* isolates show significant differences in sensitivity to fludioxonil, which may explain the lack of field performance against strawberry crown rot [26]. The results showed that eight fungicides had significantly different inhibitory effects on the mycelial growth of nine kinds of anthracnose pathogens, with prochloraz and chlorothalonil demonstrating the broadest applicability and high inhibitory effects on all nine kinds and 26 strains, followed by thiophanate and thiophanate-methyl [27]. To determine the efficacy of fungicides against mango anthracnose caused by *Colletotrichum gloeosporioides*, several fungicides were evaluated. Disease index is a quantitative measure used to assess the severity or prevalence of a disease within a crop or plant population. The disease index provides a standardized method to compare the effectiveness of different fungicides and is crucial for evaluating their efficacy in disease management. Among the tested fungicides, the combination of trifloxystrobin and tebuconazole at a concentration of 0.04% demonstrated the lowest disease index of 6.78% and the highest fruit yield, amounting to 106 kg per plant. Subsequently, the azoxystrobin and difenoconazole mixture, also at 0.04%, exhibited a disease index of 8.23% and a fruit yield of 102 kg per plant [28]. In the previous study, 75% chlorothalonil and 50% thiophanate methyl were identified as the most effective fungicides against anthracnose fungus in tea trees at a tested concentration of 5 μg/mL. At a concentration of 50 μg/mL, 10% benzothiazole and 250 g/L propiconazole also showed strong inhibition rates. However, compared to the top performers, the effectiveness of other tested fungicides, such as pyrazole ether, nitrile, mancozeb, azoxystrobin, and fludioxonil, gradually decreased. These findings can be utilized to guide the selection of appropriate fungicides for managing anthracnose in tea trees [29]. Indoor toxicity testing on *C. kahawae* revealed that 70% thiophanate methyl and 30% tebuconazole can completely inhibit the growth of *C. gloeosporioides.* [30]. The germination of conidia from both *Leptosphaeria maculans* and *L. biglobosa* was affected by fungicides, with the greatest impact observed on *L. maculans* conidia. Both triazole fungicides, flusilazole and tebuconazole, completely inhibited the germination of *L. maculans* conidia and reduced the germination of *L. biglobosa* conidia [31]. However, toxicity testing of fungicides against anthracnose on mulberry trees caused by *C. fructicola* in Zhejiang Province has not yet been reported. Consequently, it is necessary to screen and determine a highly effective fungicide for *C. fructicola*.

Anthracnose poses a significant threat to the growth of mulberry trees and the development of the sericulture economy. This study employed the cross method and spore counting method to investigate the biological characteristics of *C. fructicola*. The sensitivity of the pathogen to seven fungicides was determined using the growth rate method. The findings provide a theoretical basis and effective means for the prevention and control of mulberry tree anthracnose disease.

## 2. Materials and Methods

### 2.1. Pathogenic Fungus Strains, Growth Conditions and Fungicides

The pathogenic fungus used in this study was *C. fructicola* Cm-ZJ-1 from the Mulberry Tree Protection Laboratory, Jiangsu University of Science and Technology. The samples were isolated from diseased mulberry trees in Qiandao Lake, Zhejiang, China, in July 2023. The Cm-ZJ-1 strain was cultured on potato dextrose agar (PDA, 20% diced potato, 2% glucose, and 1.5% agar in distilled water) plates and incubated at 25 ± 2 °C for 7 days. All fungicides tested in this experiment are commercially available products sold in the market, and the specific information is provided in Table 1.

### 2.2. Effects of Different Media on the Growth of C. fructicola

To study the mycelial growth and conidia production of *C. fructicola* strain Cm-ZJ-1 on different media, eight media types were selected for experimentation. These included MS (Murashige and Skoog) medium, PDA (potato dextrose agar) medium, OA (oatmeal agar) medium, SA (starch agar) medium, CPA (carbohydrate peptone agar) medium, RB (red Bengal agar) medium, MA (malt agar) medium, and YEA (yeast extract agar) medium. Mycelial plugs (5 mm) from the single-spore cultures were transferred from the edge of a seven-day-old colonies and placed in the center of Petri dishes containing fresh test medium. The cultures were incubated in darkness at 28 °C. Mycelial growth was recorded on the 5th day post-inoculation, and the diameter of each colony was measured using the criss-cross method. Each experiment was repeated three times [32]. Meanwhile, after 10 days of cultivation, ten agar–mycelium plugs (5 mm diameter) were taken from the same position on each of the different culture media. These mycelial plugs were added to 8 mL of sterile water, thoroughly mixed, and then filtered through a double-layer of gauze. The solution was subsequently diluted, and a hemocytometer was used to count the number of conidia per square centimeter of the colony, calculated using the following formula [33].
Conidia production (conidia/cm^2^) = X × N × 5 × 10^4^/nr^2^π

X: dilution fold, N: the number of conidia in five squares of the hemacytometer, n: the number of perforated agar-mycelium plugs, and r: the inner diameter of the perforator [34].

### 2.3. Effects of Different Temperatures on the Growth of C. fructicola

The influence of temperature on mycelium growth and conidia production was studied using PDA medium, with incubation temperatures ranging from 5 to 40 °C at 5 °C intervals. A seven-day-old colony was placed in the center of a Petri dish containing PDA medium and then incubated at different temperatures. The mycelial growth was recorded 5 days post-inoculation. Conidia production in PDA medium under different temperatures was calculated [35]. The experiment was repeated three times.

### 2.4. Effect of pH on Colony Growth and Sporulation of Mulberry Anthracnose

The pH value of the PDA medium was adjusted to 4, 5, 6, 7, 8, 9, 10, 11, and 12 using 0.1% NaOH solution and 0.1% HCl solution (*V*/*V*), respectively. Mycelial plugs (5 mm) from single-spore cultures were transferred to the center of the PDA medium with different pH levels. Each treatment was repeated three times [36].

### 2.5. Effect of Photoperiod on Colony Growth and Sporulation of Mulberry Anthracnose

Under the aseptic conditions, mycelial plugs from the edge of the tested colony were transferred to the center of PDA plates. The plates were then cultured under three different lighting conditions: 24 h of continuous darkness, 12 h of light followed by 12 h of darkness, and 24 h of continuous light. Each treatment was repeated three times.

### 2.6. Determination of Mycelial Lethal Temperature of C. fructicola

The mycelial plugs (5 mm) were picked and placed into a sterilized centrifuge tube containing 2 mL of sterile water. The tubes were then placed in a constant temperature water bath preheated to temperatures ranging from 35 to 55 °C (with a gradient of 5 °C) for 10 min (preheated for 1 min). Afterward, the plugs were removed and cooled to room temperature [37]. The treated *C. fructicola* were placed in the center of PDA medium plates, and each treatment was repeated three times. The growth status of the treated *C. fructicola* was observed continuously for 7 days, with 28 °C serving as the control (CK). The lethal temperature range was determined, and within this range, the temperature gradient was set to 1 °C to repeat the above experiments to determine the precise lethal temperature.

### 2.7. Screening of Indoor Fungicides for C. fructicola

Referring to the fungicides with high efficiency and low intensity in the control of other plant anthracnose, seven fungicides were selected for preliminary screening of *C. fructicola* Cm-ZJ-1 (Table 2). These fungicides were purchased from a local distributor and diluted to various concentrations for further experiments (Table 1). The final concentrations were as follows: difenoconazole prochloraz, difenoconazole, and carbendazim at 0.625, 1.250, 2.500, 5.000, and 10.000 μg/mL; thiophanate-methyl at 0.750, 1.500, 3.000, 6.000, and 12.000 μg/mL; and fludioxonil, azoxystrobin, and pyraclostrobin at 2.500, 5.000, 10.000, 20.000, and 40.000 μg/mL, respectively. The indoor virulence of mulberry anthracnose was determined using the growth rate method. To prepare the fungicides plates, 1 mL of each solution (50 times the final concentration) was added to 49 mL of sterilized PDA medium at 50 °C, mixed well, and poured into sterilized 9 cm plates. Mycelial plugs (5 mm) were punched from the edge of an actively growing *C. fructicola* colony and inoculated onto the PDA plates with different concentrations of fungicides. The blank PDA plate without fungicides was used as a control. Each treatment was replicated four times. The plates were incubated in the dark at 25 °C. After 7 days, the colony diameter of each treatment was measured. The following equation was used to determine the inhibition efficiency of the fungicides, and EC_50_ (concentration for 50% of maximal effect) values were calculated [33].
$$R\;(\%)=\;\frac{\left(C-T\right)}{\left(C-F\right)}\times 100\%$$

R: relative inhibition rates, C: diameter of the fungus in the control, T: diameter of the fungus in the treatment, and F: diameter of fungus plugs [38].

The virulence regression equations were fitted using the common logarithm (log x) of the fungicide mass concentration as the independent variable and the inhibition rate (y) as the dependent variable. The effective concentration (EC_50_) at which each fungicide inhibits *C. fructicola* by 50% was calculated using IBM SPSS analytics [39]. This enabled a comparison of the virulence of the tested fungicides against anthracnose. Each fungicide treatment and control contained three replicate plates, and the experiment was performed twice.

### 2.8. The Effect of Fungicides on the Mycelial Morphology of C. fructicola

To observe the effect of fungicides on mycelial morphology, a wet chamber culture method with glass slides was used. Sterile filter paper was laid at the bottom of sterile culture dishes, and a U-shaped glass rod was placed in the center of each dish to create a culture chamber. Subsequently, under aseptic conditions, a small amount of PDA thin-layer culture media containing various fungicides was carefully poured into the culture dish to form the media layer. Using a sterile scalpel, a thin slice of the culture medium with a side length of approximately 1 cm was cut and placed in the center of a sterile glass slide. A small amount of *C. fructicola* spores was collected using a sterile inoculation loop and inoculated around the agar block on the glass slide, which was then covered with a coverslip for cultivation. Finally, the glass slide was positioned on the U-shaped glass rod in the center of the culture dish, and the filter paper at the bottom was moistened with 50% glycerol. *C. fructicola* grew horizontally along the coverslip, between the glass slide and the cover glass. After 7 days of cultivation, the samples were observed under an optical microscope.

### 2.9. Statistical Analysis

All experiments were systematically repeated at least three times, consistently yielding similar results. The experimental data were analyzed for variance and means using Excel software, version 2021, and the EC_50_ values of different fungicides were calculated using the IBM SPSS Statistics software, version 26.0 (IBM Corp., Armonk, NY, USA). Significant differences (*p* < 0.05) were performed using Duncan’s new multiple range test. Graphical representations were created using OriginLab origin software, version 2021.

## 3. Results

### 3.1. The Effect of Culture Medium on the Growth and Sporulation of C. fructicola

In order to assess the influence of diverse media on the growth and sporulation capabilities of *C. fructicola*, a 7-day cultivation experiment was conducted under identical conditions. The research results indicate that there are significant differences in the growth and sporulation of *C. fructicola* on different media (Figure 1). The mycelium growth rate was fastest on PDA and SA, with colony diameters significantly larger than those observed in other treatments. Although mycelium growth was also rapid on OA, CPA, and YEA, the colony diameters were slightly smaller compared to those on PDA and SA. Notably, the growth rate on MS, MA, and RB was slower, resulting in significantly smaller colony diameters than those of other treatments. Particularly, RB medium proved to be unsupportive of colony growth, showing the smallest colony diameter among all treatments. In terms of sporulation, PDA stood out as the optimal medium for the highest yield of spores by *C. fructicola*. Conversely, OA, YEA, SA, CPA, MS, and RB exhibited reduced spore production. Notably, MA did not support any spore production at all (Figure 1).

### 3.2. Effects of Different Temperatures on Colony Growth and Sporulation of C. fructicola

To comprehensively assess the impact of varying temperature conditions on the growth and sporulation capacity of *C. fructicola*, we conducted a series of measurements on PDA plates at temperatures ranging from 5, 10, 15, 20, 25, 30, 35, to 40 °C, analyzing both the growth rate and sporulation ability under these conditions (Figure 2). The research results indicate that temperature had a significant effect on the colony growth and spore production of *C. fructicola.* Specifically, optimal growth conditions for the mycelium were observed within a temperature range of 20 °C to 35 °C, with peak growth velocity and colony diameter achieved at 25 °C, significantly surpassing those observed in other temperature treatments (Figure 2). Conversely, growth was notably sluggish at 15 °C and 40 °C, and virtually absent at 5 °C and 10 °C, indicating inhibitory effects at these extreme temperatures. Regarding sporulation, the findings reveal a pronounced peak in spore production within the temperature range of 20 °C to 30 °C, with maximum spore yield recorded at 25 °C, significantly exceeding that of all other tested temperatures. Sporulation was reduced at 10 °C, 15 °C, and 35 °C, while it was minimal at 40 °C and completely absent at 5 °C, further emphasizing the sensitivity of *C. fructicola* sporulation ability to temperature (Figure 2).

### 3.3. The Effect of Different pH on Colony Growth and Sporulation of C. fructicola

To evaluate the influence of pH on the growth and sporulation capacities of *C. fructicola*, an experimental study was conducted. In this study, the mycelial growth and spore production of *C. fructicola* were quantitatively assessed on potato dextrose agar (PDA) media adjusted to different pH levels (Figure 3). The strain is capable of growing and producing spores across a pH range from 4 to 12. Optimal growth and sporulation conditions for the strain are observed between pH 6 and 9, with peak performance occurring at a pH of 7 (Figure 3). Conversely, the strain exhibits suboptimal growth and sporulation at pH levels of 4 to 5 and 10 to 12, indicating a preference for neutral to slightly acidic pH environments for growth and spore production (Figure 3).

### 3.4. The Effect of Light Periods on Colony Growth and Sporulation of C. fructicola

Visible light can trigger a molecular pathway, generating an oxidative stress response that leads to repressed mycelial growth in some fungal species, and it also serves as a signal for potentially more harmful ultraviolet (UV) light [40]. This study evaluated the effects of different light conditions on *C. fructicola* from mulberry anthracnose on potato dextrose agar (PDA) media. The results showed that there were significant differences in colony diameter and spore production of *C. fructicola* under different photoperiods (Figure 4). A 12 h/12 h (light/dark) alternating cycle was found to be favorable for both the growth and sporulation of *C. fructicola*, with results significantly higher compared to the other tested photoperiods (Figure 4). Under continuous 24 h light or darkness, mycelial growth was slower, and sporulation was reduced, with no significant difference observed between these two conditions (Figure 4).

### 3.5. Determination of the Lethal Temperature for Mycelia of Mulberry Anthracnose

The fungal cake of the mulberry anthracnose fungus Cm-ZJ-1 was subjected to water baths at varying temperatures for experimentation. The results indicated that mycelial plugs could grow and produce colonies when soaked in water at temperatures ranging from 35 to 55 °C for 10 min. However, they did not grow when exposed to temperatures of 60 °C and above. To further refine the temperature threshold, the gradient was narrowed to 1 °C between 55 and 60 °C for repeated experiments. These experiments revealed that the fungal cake could not grow normally after being soaked in water at 56 °C for 10 min, indicating that the lethal temperature for the fungus is 56 °C for 10 min.

### 3.6. Comparison of Indoor Virulence of Several Fungicides Against Mulberry Anthracnose

This study conducted indoor toxicity tests on seven commonly used fungicides using the growth rate method. The results of the indoor toxicity tests showed that all seven fungicides exhibited inhibitory effects on *C. fructicola*. The inhibitory effect increased with rising fungicide concentrations, and there were significant differences in the inhibitory effects among the different fungicides (Figure 5, Table 2). In this study, when the concentrations of difenoconazole prochloraz, difenoconazole, and carbendazim were in the range of 0.625–10.000 μg/mL, the inhibition rates on the mycelial growth of *C. fructicola* were 53.70–87.19%, 39.04–65.12%, and 67.13–79.32%, respectively (Figure 5A–C). For thiophanate-methyl, at concentrations ranging from 0.750 to 12.000 μg/mL, the inhibition rate on the mycelial growth of *C. fructicola* was 27.16–63.12% (Figure 5G). Additionally, when fludioxonil was at concentrations of 2.500–40.000 μg/mL, the inhibition rate on the mycelial growth of *C. fructicola* Cm-ZJ-1 ranged from 11.11% to 37.35% (Figure 5D). For pyraclostrobin, at concentrations of 2.500–40.000 μg/mL, the inhibition rate on the mycelial growth of *C. fructicola* was 39.51–74.38% (Figure 5E). Lastly, for azoxystrobin, at concentrations of 2.500–40.000 μg/mL, the inhibition rate on the mycelial growth of *C. fructicola* was 26.39–44.44% (Figure 5F, Table 2). The correlation coefficients of the toxicity regression equations for the seven fungicides range from 0.7 to 1.0, indicating a strong correlation between the dosage of the fungicides and their inhibitory effects (Table 2). There are significant differences in the toxicity of the seven fungicides against the mulberry anthracnose Cm-ZJ-1. The fungicide with the best inhibitory effect is 97% carbendazim WP, which exhibits the strongest bactericidal activity with an EC_50_ of 0.0242 μg/mL (Table 3). This is followed by 35% difenoconazole prochloraz AEC, with an EC_50_ of 0.4180 μg/mL, and then 25% difenoconazole EC, with an EC_50_ of 2.2350 μg/mL (Table 3). The EC_50_ values of the remaining four fungicides are higher, indicating weaker bactericidal effects. Notably, the bactericidal effect of 25 g/L fludioxonil SD is relatively poor, with an EC_50_ value of 103.0170 μg/mL (Table 3).

### 3.7. Effects of Fungicides on the Mycelial Morphology of Mulberry Anthracnose

An experiment was conducted to investigate the effects of various fungicides on the growth morphology of mulberry anthracnose Cm-ZJ-1 using the wet room culture method with glass slides. By observing mycelial growth under an optical microscope, the results showed significant differences between the control group and the treatment groups exposed to different fungicides (Figure 6). In the control group, the mycelium exhibited typical growth characteristics, featuring uniform thickness, elongated branches, and a dense covering of conidia at the tips (Figure 6H). Conversely, the application of carbendazim resulted in a noticeable reduction in mycelial density, with a loose and sparse arrangement observed (Figure 6A). Upon treatment with difenoconazole prochloraz, the mycelial growth pattern of the fungus approached normality, albeit with a more dispersed distribution (Figure 6B). In contrast, difenoconazole induced sparsity and a loose arrangement in the hyphae, accompanied by entanglement deformities (Figure 6C). Thiophanate-methyl treatment led to a distinct phenotype, where the fungal hyphae were densely packed and appeared fused together (Figure 6D). Similarly, pyraclostrobin fungicide caused the hyphae to become sparse and shortened, with increased irregular branching, intertwining, and an overall irregular arrangement (Figure 6E). Azoxystrobin, on the other hand, triggered a response where the fungal hyphae became densely congregated, partially fusing together and exhibiting deformation (Figure 6F). Upon treatment with fludioxonil, the hyphae were sparse and loosely arranged (Figure 6G). These findings are consistent with the results of indoor toxicity assessments, indicating that all seven fungicides can affect the normal growth and development of *C. fructicola*. Notably, there are significant differences in the extent of mycelial deformity caused by various fungicides.

## 4. Discussion

*Colletotrichum* is one of the most common and important genera of plant pathogenic fungi. Virtually every crop grown throughout the world is susceptible to one or more species of *Colletotrichum* [41,42]. Anthracnose is an important plant pathogen that can infect various plants and cause destructive diseases worldwide and seriously affecting the development of global agricultural production [43,44]. At present, the prevention and control of anthracnose still primarily rely on chemical fungicides, and this approach is expected to continue for a long time in the future [45,46,47]. Numerous studies have been conducted on the prevention and control of anthracnose in crops such as pear trees, peppercorns, chili peppers, peach trees, and citrus. The prevention and control effects of various fungicides on different crops are inconsistent, which may be related to the different types of pathogens and their varying sensitivities to chemical fungicides [48,49,50].

Leaf spot disease is one of the most common and highly prevalent fungal diseases in mulberry production in Zhejiang Province, which seriously affects the yield and quality of mulberry leaves and causes significant economic losses. Our research group previously identified the pathogen causing mulberry leaf spot disease in the area and confirmed that the pathogen responsible for the disease was *Colletotrichum fructicola*. This finding is inconsistent with the results reported for anthracnose in Sichuan mulberry trees, where *C. fructicola* and *C. cliviae* have been reported as endophytic bacteria rather than pathogenic bacteria in mulberry trees [16]. The pathogen causing anthracnose in mulberry trees in Wuhan has been identified as *C. aenigma*, which further illustrates the complexity and diversity of anthracnose fungal species [17]. The pathogen causing anthracnose in mulberry trees in Baiwangshan Forest Park, Beijing, China, has been identified as *C. gloeosporioides* [15]. However, the strain Cm-ZJ-1 used in this study is the pathogenic bacterium of mulberry anthracnose in Zhejiang Province, which further indicates that the pathogens causing mulberry anthracnose in different regions are not consistent. It is speculated that due to differences in ecosystems in different regions, the same disease may be caused by different types of pathogens [44]. This ecological diversity reflects the complexity and diversity of microorganisms in nature. The anthracnose strain Cm-ZJ-1 in this study exhibits strong pathogenicity and a high isolation frequency, and is the main pathogenic bacterium of mulberry anthracnose in the Qiandao Lake area of Zhejiang Province. This differs from the previously reported main pathogenic bacteria of mulberry anthracnose.

This study determined the biological characteristics of mulberry anthracnose and found that different cultivation conditions can affect the growth and sporulation of the *C. fructicola* (Figure 1, Figure 2, Figure 3 and Figure 4). The results showed that the strain could grow normally on the tested media and could produce spores on the media except MA. The mycelium on PDA and SA grew faster, and the sporulation was the most on PDA and OA (Figure 1). The pathogen has a wide range of adaptation to temperature, and can grow at 5–40 °C, with the optimum temperature is 25 °C, but the colony hardly grows at 5 °C (Figure 2). This was similar to other pathogenic *Colletotrichum* spp., such as *C. truncatum* and *C. camelliae*, which have optimum mycelial growth temperatures ranging from 25 to 30 °C [51]. This range matches the average temperature during the growing season in Hangzhou. The pathogen demonstrated strong adaptability to the pH of the culture medium, capable of growing and producing spores within a pH range of 4 to 12, with an optimal pH value of 7 (Figure 3). However, excessive acidity or alkalinity was not conducive to mycelial growth and sporulation, which was consistent with the results on the mycelial growth of spinach leaf spot diseases [23]. The strain can grow and produce spores normally under three different photoperiods. However, the optimal culture conditions are alternating light and dark cycles (12 h/12 h) (Figure 4), which align with the local photoperiod conditions of Hangzhou City, Zhejiang Province, China. This indicates that the strain does not require high light levels, which was consistent with the results of *Bipolaris bicolor* in *Momordica charantia* leaf spot disease [40]. The lethal temperature of the pathogen was 56 °C and treated with water bath for 10 min.

In this study, we conducted an indoor screening test to evaluate the inhibitory effects of seven fungicides on the growth of *C. fructicola*, the causal agent of anthracnose in mulberry. The results demonstrated that all seven fungicides exhibited inhibitory effects on *C. fructicola* growth, with the degree of inhibition increasing with the concentration of the fungicides (Figure 5). The experiment showed that 97% carbendazim WP had the best inhibitory effect among the tested fungicides, with an EC_50_ of 0.0242 μg/mL (Figure 5C, Table 3). This indicates that benzimidazole fungicides have a strong inhibitory effect on *C. fructicola*, consistent with the results for olive anthracnose caused by *C. acutatum* [50]. Sensitivity testing was performed on the fungicides mancozeb, thiophanate-methyl, and azoxystrobin against the *Colletotrichum acutatum* population prevalent in apple orchards in Brazil. The results revealed that apple trees from different parts of the plant and of various varieties displayed varying degrees of sensitivity to these fungicides [52]. Difenoconazole prochloraz, an imidazole fungicide, exerts a potent bacteriostatic effect by inhibiting ergosterol biosynthesis. A strong inhibitory effect on *C. fructicola* was demonstrated, with an EC_50_ of 0.4180 μg/mL (Figure 5A, Table 3). These findings are consistent with the sensitivity of imidazole to walnut anthracnose caused by *C. gloeosporioides* [53]. Additionally, 25% difenoconazole, 500 g/L thiophanate-methyl, and 25% pyraclostrobin fungicide exhibited strong toxicity against *C. fructicola*, with EC_50_ values of 2.2350, 4.7138, and 5.7946 μg/mL, respectively (Figure 5B,E,G, Table 3). These findings are consistent with the sensitivity results of rubber tree anthracnose pathogens to these fungicides [21]. However, the control efficacy of the 250 g/L azoxystrobin suspension and 25 g/L fludioxonil among the fungicides employed in this experiment was suboptimal, exhibiting EC_50_ values of 70.1499 μg/mL and 103.0170 μg/mL, respectively (Figure 5D,F, Table 3). The sensitivity results of grape and peach anthracnose in response to fungicides were aligned with those findings [54,55,56]. Additionally, the effect of fungicides on the morphological growth of *C. fructicola* mycelium was determined. The results showed that all tested fungicides affected the normal growth and development of *C. fructicola* mycelium as well (Figure 6). However, its mechanism is still unclear and requires further research in the future.

In this study, the biological characteristics of the mulberry leaf spot pathogen *C. fructicola* in Zhejiang Province were analyzed, and seven kinds of fungicides were selected for indoor virulence testing. The results of this study can provide a reference for the comprehensive control of mulberry tree leaf spot in Zhejiang. However, given that the fungicide test was conducted in an indoor setting, additional field trials are necessary to confirm the effectiveness of these fungicides in controlling mulberry leaf spot in Zhejiang. Overall, the main findings from our study could contribute to developing future sustainable and effective management strategies to control mulberry anthracnose disease caused by *C. fructicola*, especially in Qiandao Lake, Hangzhou, China.

## 5. Conclusions

According to the findings of our study, we have identified the biological characteristics of *Colletotrichum fructicola*, the pathogen responsible for mulberry leaf spot disease in Zhejiang Province. This study found that different cultivation conditions can affect the growth and sporulation of mulberry anthracnose (Figure 1, Figure 2, Figure 3 and Figure 4). Additionally, the indoor toxicity analysis revealed that 97% carbendazim WP and 35% difenoconazole prochloraz AEC exhibited significant inhibitory effects on mulberry anthracnose (Figure 5A,C and Figure 6A,B). There were significant differences in the extent of mycelial deformity caused by the various fungicides. This study provides a reference for the comprehensive prevention and control of mulberry leaf spot disease in Zhejiang Province.

## Figures and Tables

**Figure 1 microorganisms-12-02386-f001:**
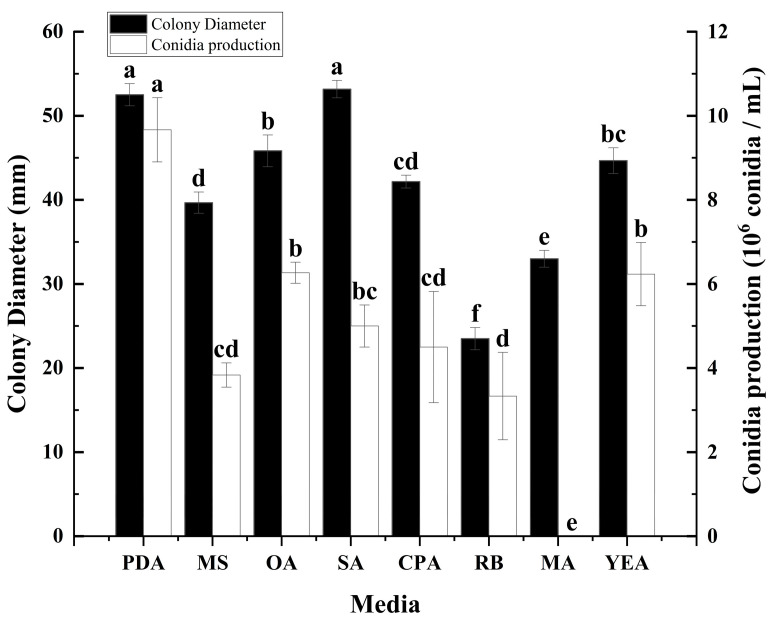
Effects of culture media on mycelial growth and sporulation of *Colletotrichum fructicola* Cm-ZJ-1. Different lowercase letters represent significant difference (*p* < 0.05). Data are mean ± SE.

**Figure 2 microorganisms-12-02386-f002:**
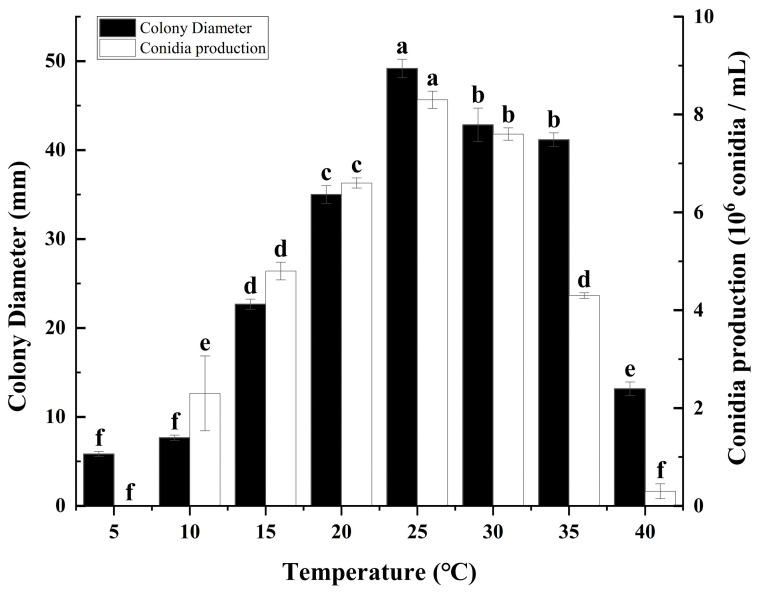
Effects of temperature on mycelial growth and sporulation of *Colletotrichum fructicola* Cm-ZJ-1. Different lowercase letters represent significant difference (*p* < 0.05). Data are mean ± SE.

**Figure 3 microorganisms-12-02386-f003:**
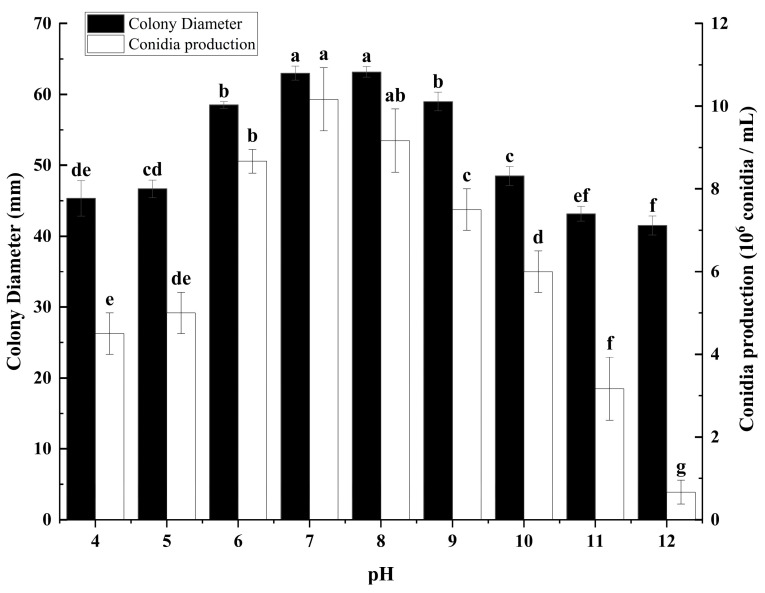
Effects of pH on mycelial growth and sporulation of *Colletotrichum fructicola* Cm-ZJ-1. Different lowercase letters represent significant difference (*p* < 0.05). Data are mean ± SE.

**Figure 4 microorganisms-12-02386-f004:**
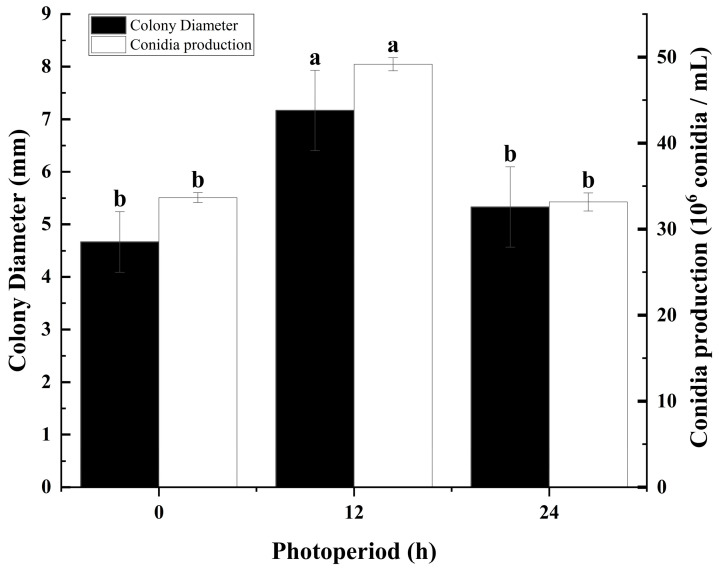
Effects of photoperiod on mycelial growth and sporulation of *Colletotrichum fructicola* Cm-ZJ-1: 0 h, 24 h in the dark; 12 h, 12 h light + 12 h dark; 24 h, all day under light exposure. Different lowercase letters represent significant difference (*p* < 0.05). Data are mean ± SE.

**Figure 5 microorganisms-12-02386-f005:**
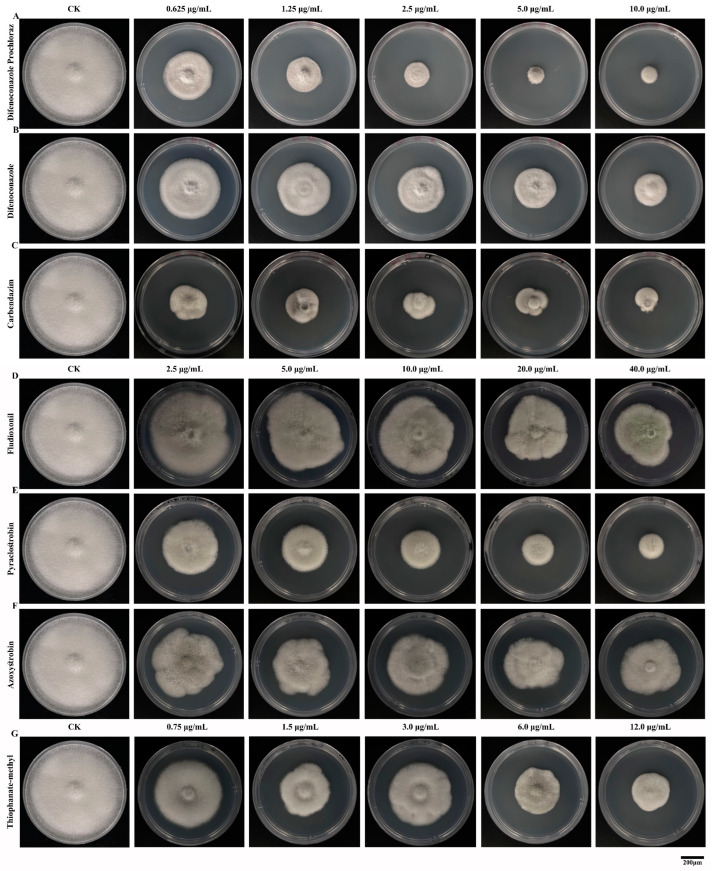
Effects of seven fungicides on the mycelial growth of *Colletotrichum fructicola* Cm-ZJ-1. Mycelial growth was assessed on PDA plates after 7 days of incubation in either the absence (CK) or presence of varying concentrations of fungicides. (**A**): Difenoconazole Prochloraz (final concentrations: 0.625, 1.250, 2.500, 5.000, and 10.000 μg/mL); (**B**): Difenoconazole (final concentrations: 0.625, 1.250, 2.500, 5.000, and 10.000 μg/mL); (**C**): Carbendazim (final concentrations: 0.625, 1.250, 2.500, 5.000, and 10.000 μg/mL); (**D**): Fludioxonil (final concentrations: 2.500, 5.000, 10.000, 20.000, and 40.000 μg/mL); (**E**): Pyraclostrobin (final concentrations: 2.500, 5.000, 10.000, 20.000, and 40.000 μg/mL); (**F**): Azoxystrobin (final concentrations: 2.500, 5.000, 10.000, 20.000, and 40.000 μg/mL); (**G**): Thiophanate-methyl (final concentrations: 0.750, 1.500, 3.000, 6.000, and 12.000 μg/mL). CK represents the control group without fungicides. Scale bars = 200 μm.

**Figure 6 microorganisms-12-02386-f006:**
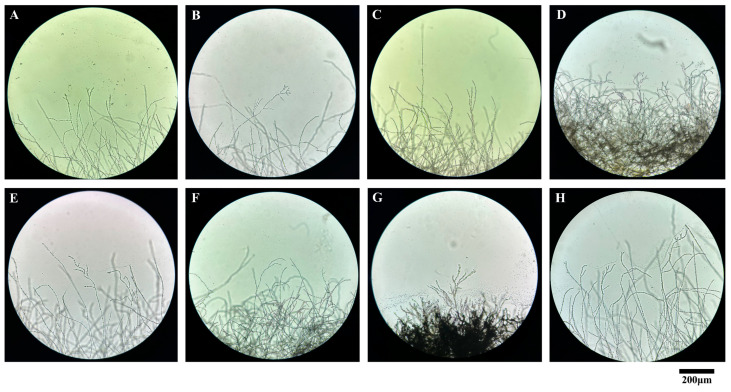
Influence of different fungicides on mycelial morphology of *C. fructicola* Cm-ZJ-1. Mycelial growth of *C. fructicola* Cm-ZJ-1 was observed on PDA plates incubated for 7 days, in either the absence of fungicides (CK) or fungicide with the highest concentration. (**A**): Carbendazim (10.000 μg/mL); (**B**): Difenoconazole Prochloraz (10.000 μg/mL); (**C**): Difenoconazole (10.000 μg/mL); (**D**): Thiophanate-methyl (12.000 μg/mL); (**E**): Pyraclostrobin (40.000 μg/mL); (**F**): Azoxystrobin (40.000 μg/mL); (**G**): Fludioxonil (40.000 μg/mL); (**H**): CK represents the control group without fungicides. The images were obtained with a microscope. Scale bars = 200 µm.

**Table 1 microorganisms-12-02386-t001:** Common names, contents, and manufacturers of seven fungicides. AEC, aqueous emulsion concentrate; EC, emulsifiable concentrate; WP, wettable powder; SC, suspension concentrate; SSCC, suspension seed coating concentrate.

Fungicide	Manufacturer	Concentration (μg/mL)
35% Difenoconazole Prochloraz AEC	Qingdao Hansheng Biological Technology Co., Ltd. (Qingdao, China)	0.625, 1.250, 2.500, 5.000, 10.000
25% Difenoconazole EC	Shenzhen Noposion Agrochemical Co., Ltd. (Shenzhen, China)	0.625, 1.250, 2.500, 5.000, 10.000
97% Carbendazim WP	Shanghai Aladdin Biochemical Technology Co., Ltd. (Shanghai, China)	0.625, 1.250, 2.500, 5.000, 10.000
500 g·L^−1^ Thiophanate-Methyl SC	Jiangsu Longdeng Chemical Co., Ltd. (Lianyungang, China)	0.750, 1.500, 3.000, 6.000, 12.000
25 g·L^−1^ Fludioxonil SSCC	Xianzhengda Nantong Crop Protection Co., Ltd. (Nantong, China)	2.500, 5.000, 10.000, 20.000, 40.000
25% Pyraclostrobin SC	Jinan Luba Pesticide Co., Ltd. (Jinan, China)	2.500, 5.000, 10.000, 20.000, 40.000
250 g·L^−1^ Azoxystrobin SC	Syngenta, (Basel, Switzerland)	2.500, 5.000, 10.000, 20.000, 40.000

**Table 2 microorganisms-12-02386-t002:** Determination of inhibition rates of seven fungicides on *C. fructicola* Cm-ZJ-1.

Fungicide Name	Concentration (μg/mL)	Inhibitory Rates (%)
35% Difenoconazole Prochloraz AEC	10.000	87.19
	5.000	86.88
	2.500	79.63
	1.250	66.20
	0.625	53.70
	CK	-
25% Difenoconazole EC	10.000	65.12
	5.000	54.63
	2.500	48.46
	1.250	47.38
	0.625	39.04
	CK	-
97% Carbendazim WP	10.000	79.32
	5.000	71.91
	2.500	75.62
	1.250	66.98
	0.625	67.13
	CK	-
500 g·L^−1^ Thiophanate-Methyl SC	12.000	63.12
	6.000	52.47
	3.000	46.30
	1.500	31.17
	0.750	27.16
	CK	-
25 g·L^−1^ Fludioxonil SSCC	40.000	37.35
	20.000	29.94
	10.000	22.80
	5.000	16.05
	2.500	11.11
	CK	-
25% Pyraclostrobin SC	40.000	74.38
	20.000	66.05
	10.000	60.19
	5.000	45.06
	2.500	49.51
	CK	-
250 g·L^−1^ Azoxystrobin SC	40.000	44.44
	20.000	42.90
	10.000	38.27
	5.000	38.58
	2.500	26.39
	CK	-

**Table 3 microorganisms-12-02386-t003:** Inhibitory effect of seven fungicides on *C. fructicola* Cm-ZJ-1.

Fungicide Names	Regression Equation	R^2^	EC_50_/(μg·L^−1^)
35% Difenoconazole Prochloraz AEC	y = 0.9261× + 5.3506	0.9370	0.4180
25% Difenoconazole EC	y = 0.5035× + 4.8241	0.9386	2.2350
97% Carbendazim WP	y = 0.2954× + 5.4774	0.7405	0.0242
500 g·L^−1^ Thiophanate-Methyl SC	y = 0.8101× + 4.4545	0.9767	4.7138
25 g·L^−1^ Fludioxonil SSCC	y = 0.7515× + 3.4873	0.9988	103.0170
25% Pyraclostrobin SC	y = 0.7908× + 4.3966	0.9808	5.7946
250 g·L^−1^ Azoxystrobin SC	y = 0.3637× + 4.3286	0.7991	70.1499

## Data Availability

The data presented in this study are available within the article.

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
