# Peer review of "Biological Characteristics and Fungicide Screening of *Colletotrichum fructicola* Causing Mulberry Anthracnose"

_microorganisms, 2024, doi:10.3390/microorganisms12122386_

Round 1
Reviewer 1 Report
Comments and Suggestions for Authors
Biological Characteristics and Fungicide Screening of Colletotrichum fructicola Causing Mulberry Anthracnose
Abstract
Use as key words significant words for the study but not the ones in title.
Introduction
OK
Methodology
No table 1 is included.
Results, discussion
Ok
Conclusion
OK
See comments in MS

Reviewer 2 Report
Comments and Suggestions for Authors
Li et al. reported the biological characteristics of a Mulberry anthracnose causing fungus, Colletotrichum fructicola and screened it against various fungicide. They systematically studied the growth conditions for this fungi including media, temperature and pH which generates information that the research community can use for reference. The introduction is well written and provides the details about the importance of Mulberry, severity of anthracnose as well as various fungicides available to treat the condition. The experiments are well designed, explained well and provide scientific rigor to validate the research objectives. Following are some comments that I would like the authors to address:
1. Page 2 line 56: It is not clear what the authors meant by “overwinter”.
2. The authors mentioned there are 9 kinds and 26 strains of anthracnose pathogens in line 113. Can the authors clarify what are these 9 kinds?
3. Can the authors briefly mention what a disease index is and why it is important in testing fungicide efficacy before they discuss this in line 117?
4. On section 2.5, the authors stated that the plates were cultured under 4 different conditions but then lists 3 different conditions. Which one is correct? Please correct.
5. Can the authors clarify what do the letters a/b/c etc. mean on figure 1?
6. Please include in figure 2 and 3 captions that this is on PDA plate, also on figure 4 if the PDA plates are also used in light exposure experiment.
Author Response
Response to Reviewer 2 comments:
Thank you very much for your thorough review of our manuscript and for the valuable comments you have provided. We greatly appreciate the detailed and constructive feedback provided by the reviewers. Below, we address each of the reviewer's comments and provide the necessary revisions and explanations.
Comment 1: Page 2 line 56: It is not clear what the authors meant by “overwinter”.
Response: Thanks for the reviewer’s comments and suggestions. We are very sorry for the unclear expression. On Page 2, line 56, "Overwinter" refers to the ability of an organism, such as C. dematium, to survive the winter season, typically within infected or latently infected leaves, and serve as a primary source of infection in the following year when conditions become favorable. We have carefully refined the sentence structure to enhance clarity in the revised version.
Comment 2: The authors mentioned there are 9 kinds and 26 strains of anthracnose pathogens in line 113. Can the authors clarify what are these 9 kinds?
Response: Thanks for the reviewer’s comments. We appreciate your interest in the nine kinds of anthracnose pathogens mentioned in line 113. Here, we provide the detailed list of these pathogens: C. fioriniae, C. fructicola, C. siamense, C. nymphaeae, C. americae-borealis, C. alienum, C. coelogynes, C. tamarilloi, C. liaoningense. These pathogens, which cause anthracnose in thin shelled walnuts, were collected from infected leaves and fruits across Zhejiang, Jiangxi, Anhui, and Yunnan provinces in China [27].
Comment 3: Can the authors briefly mention what a disease index is and why it is important in testing fungicide efficacy before they discuss this in line 117?
Response: Thanks for the reviewer’s comments and suggestions. We have added a brief explanation of the disease index and its importance in testing fungicide efficacy before discussing it in line 117. The revised text is as follows: Disease index is a quantitative measure used to assess the severity or prevalence of a disease within a crop or plant population. It is calculated based on the percentage of affected plants and the degree of disease severity on each plant. The disease index provides a standardized method to compare the effectiveness of different fungicides and is crucial for evaluating their efficacy in disease management.
Comment 4: On section 2.5, the authors stated that the plates were cultured under 4 different conditions but then lists 3 different conditions. Which one is correct? Please correct.
Response: Thanks for the reviewer’s comments and suggestions. Thank you for pointing out the numerical errors in section 2.5. In section 2.5, the plates were cultured under three different conditions: 0 h (24 h in the dark); 12 h (12 h light+12 h dark); 24 h (continuous light exposure). We apologize for any confusion this may have caused and appreciate your attention to detail.
Comment 5: Can the authors clarify what do the letters a/b/c etc. mean on figure 1?
Response: Thanks for the reviewer’s comments. The letters "a," "b," "c," etc., in the figure indicate statistical significance among the different media tested. Specifically, bars labeled with identical letters (e.g., "a") belong to the same group and are statistically indistinguishable from each other. Conversely, bars marked with different letters (e.g., "a" vs. "b") belong to different groups and exhibit significant differences in colony diameter. For example, both PDA and SA have the label "a" above their respective bars, indicating no significant difference in colony diameter. In contrast, RB is labeled with "f", and MA is labeled with "e", showing significant differences in colony diameter compared to most others media.
Comment 6: Please include in figure 2 and 3 captions that this is on PDA plate, also on figure 4 if the PDA plates are also used in light exposure experiment.
Response: Thanks for the reviewer’s comments. Yes. On figure 4, the PDA plates were also used in the light exposure experiment, where they were cultured under three different conditions.
